# An Evaluation of Understudied Phytocannabinoids and Their Effects in Two Neuronal Models

**DOI:** 10.3390/molecules26175352

**Published:** 2021-09-02

**Authors:** Alex Straiker, Sierra Wilson, Wesley Corey, Michaela Dvorakova, Taryn Bosquez, Joye Tracey, Caroline Wilkowski, Kathleen Ho, Jim Wager-Miller, Ken Mackie

**Affiliations:** 1Gill Center for Molecular Bioscience, Program in Neuroscience, Department of Psychological & Brain Sciences, Indiana University, Bloomington, IN 47405, USA; silywils@iu.edu (S.W.); coreyw@iu.edu (W.C.); michdvor@iu.edu (M.D.); tbosquez@iu.edu (T.B.); jxtracey@iu.edu (J.T.); cwilkows@iu.edu (C.W.); katkho@iu.edu (K.H.); jm99@indiana.edu (J.W.-M.); kmackie@iu.edu (K.M.); 2Department of Molecular Pharmacology, Institute of Molecular Genetics of the Czech Academy of Sciences, Videnska 1083, 14220 Prague, Czech Republic

**Keywords:** cannabichromene, cannabidiolic acid, cannabidivarin, cannabidivarinic acid, phytocannabinoids, tetrahydrocannabivarin

## Abstract

Cannabis contains more than 100 phytocannabinoids. Most of these remain poorly characterized, particularly in neurons. We tested a panel of five phytocannabinoids—cannabichromene (CBC), cannabidiolic acid (CBDA), cannabidivarin (CBDV), cannabidivarinic acid (CBDVA), and Δ^9^-tetrahydrocannabivarin (THCV) in two neuronal models, autaptic hippocampal neurons and dorsal root ganglion (DRG) neurons. Autaptic neurons expressed a form of CB1-dependent retrograde plasticity while DRGs expressed a variety of transient receptor potential (TRP) channels. CBC, CBDA, and CBDVA had little or no effect on neuronal cannabinoid signaling. CBDV and THCV differentially inhibited cannabinoid signaling. THCV inhibited CB1 receptors presynaptically while CBDV acted post-synaptically, perhaps by inhibiting 2-AG production. None of the compounds elicited a consistent DRG response. In summary, we find that two of five ‘minor’ phytocannabinoids tested antagonized CB1-based signaling in a neuronal model, but with very different mechanisms. Our findings highlight the diversity of potential actions of phytocannabinoids and the importance of fully evaluating these compounds in neuronal models.

## 1. Introduction

Cannabis has been used extensively during much of human history. Due to its changing legal status, cannabis and its phytocannabinoid constituents have recently attracted a great deal of commercial and public interest. Specifically, Δ^9^-tetrahydrocannabinol (Δ^9^-THC) and cannabidiol (CBD) are the most abundant (and studied) phytocannabinoids; however, more than 100 additional phytocannabinoids are present at lower concentrations in cannabis [1], the so-called “minor cannabinoids”. Δ^9^-THC, the main intoxicating component of cannabis [2], has been the subject of thousands of studies; research shows that it acts on the endocannabinoid signaling system [3]. This signaling system includes receptors, primarily CB1 [4] and CB2 [5], but also other lipid messengers (endocannabinoids) and enzymes to synthesize and metabolize these messengers ‘on demand’ [6]. CB1 receptors are enriched in the brain and likely mediate many of the CNS effects of THC. The case of CBD is perhaps more interesting. CBD, as the main non-intoxicating constituent of cannabis [7], was long considered inactive; yet, in the space of 10 years, CBD has transitioned from relatively unknown to a wonder-drug in the popular press, due in part to CBD approval as a therapy treatment for seizures associated with Dravet syndrome and Lennox–Gastaut syndrome, two forms of childhood epilepsy [8]. CBD is now readily available over-the-counter, in a variety of preparations, in grocery stores in many US states. Commercial interests have taken note of the renewed interest in phytocannabinoids and are now focusing on minor cannabinoids. Previously, low and variable concentrations of minor cannabinoids in cannabis served as natural limitations on their exploitation. However, these limitations have been overcome with scaled production and improvements in extraction and synthesis; some groups are even harnessing yeast or algae to synthesize specific cannabinoids [9]. Phytocannabinoids, such as cannabidiolic acid (CBDA) and cannabichromene (CBC), are now finding their way into creams, foods, and beverages, with vendors ascribing health benefits to these compounds, and consumers readily embracing all things cannabinoid. Though it is often assumed that phytocannabinoids act via the cannabinoid signaling system [10], there has been little systematic study on how these compounds work in the body. A first question is whether they interact with the cannabinoid signaling system. Binding studies can be misleading since CBD binds poorly to the orthosteric site of CB1 receptors [11], but effectively and potently inhibits CB1 as a negative allosteric modulator [12,13]. Moreover, the cannabinoid signaling system consists not only of cannabinoid receptors but also of lipid messengers, and the enzymes to synthesize, transport, and metabolize these messengers [14], all processes that ‘minor’ cannabinoids might affect. Research shows that phytocannabinoids activate several members of the transient receptor potential (TRP) family of ion channels [15]. Some of these receptors may be linked to the endocannabinoid signaling system, since anandamide is an efficacious agonist at TRPV1 [16]. We have therefore tested several ‘minor’ phytocannabinoids—CBC, CBDA, cannabidivarin (CBDV), cannabidivarinic acid (CBDVA), and Δ^9^-tetrahydrocannabivarin (THCV) (Figure 1), in two neuronal models of endogenous cannabinoid signaling. These phytocannabinoids were chosen for their range of reported effects (spanning from analgesia to seizures) and because they are some of the most commercially promoted compounds. The neuronal models include the well-characterized autaptic hippocampal neurons that natively express CB1 receptors, the machinery to synthesize and metabolize the endocannabinoid 2-AG, as well as several forms of CB1-mediated plasticity [17,18,19]. We also tested dorsal root ganglion neurons, which natively express a variety of TRP receptors.

## 2. Materials and Methods

### 2.1. Hippocampal Culture Preparation

All procedures used in this study were approved by the Animal Care Committee of Indiana University and conformed to the Guidelines of the National Institutes of Health on the Care and Use of Animals. Mouse hippocampal neurons isolated from the CA1–CA3 region were cultured on microislands as described previously [20,21]. Neurons were obtained from animals (age postnatal day 0–2) and plated onto a feeder layer of hippocampal astrocytes that had been laid down previously [22]. Cultures were grown in high-glucose (20 mM) DMEM containing 10% horse serum, without mitotic inhibitors, and used for recordings after 8 days in culture, and for no more than three hours after removal from culture medium.

### 2.2. Electrophysiology

When a single neuron is grown on a small island of permissive substrate, it forms synapses or “autapses” onto itself. All experiments were performed on isolated autaptic neurons. Whole cell voltage-clamp recordings from autaptic neurons were carried out at room temperature using an Axopatch 200A amplifier (Molecular Devices, Sunnyvale, CA). The extracellular solution contained (in mM) 119 NaCl, 5 KCl, 2.5 CaCl_2_, 1.5 MgCl_2_, 30 glucose, and 20 HEPES. Continuous flow of solution through the bath chamber (~2 mL/min) ensured rapid drug application and clearance. Drugs were typically prepared as stocks, and then diluted into extracellular solution at their final concentration and used on the same day.

Recording pipettes of 1.8–3 MΩ were filled with (in mM) 121.5 K Gluconate, 17.5 KCl, 9 NaCl, 1 MgCl_2_, 10 HEPES, 0.2 EGTA, 2 MgATP, and 0.5 LiGTP. Access resistance and holding current were monitored and only cells with both stable access resistance and holding current were included for data analysis. Conventional stimulus protocol: the membrane potential was held at –70 mV and excitatory postsynaptic currents (EPSCs) were evoked every 20 s by triggering an unclamped action current with a 1.0 ms depolarizing step. The resultant evoked waveform consisted of a brief stimulus artifact and a large downward spike representing inward sodium currents, followed by the slower EPSC. The size of the recorded EPSCs was calculated by integrating the evoked current to yield a charge value (in pC). Calculating the charge value in this manner yields an indirect measure of the amount of neurotransmitter released while minimizing the effects of cable distortion on currents generated far from the site of the recording electrode (the soma). Data were acquired at a sampling rate of 5 kHz.

DSE stimuli: after establishing a 10–20 s 0.5 Hz baseline, DSE was evoked by depolarizing to 0 mV for 50 ms, 100 ms, 300 ms, 500 ms, 1 s, 3 s and 10 s, followed in each case by resumption of a 0.5 Hz stimulus protocol for 20–80+ seconds, allowing EPSCs to recover to baseline values. This approach allowed us to determine the sensitivity of the synapses to DSE induction. To allow comparison, baseline values (prior to the DSE stimulus) are normalized to one. DSE inhibition values are presented as fractions of 1, i.e., a 50% inhibition from the baseline response is 0.50 ± standard error of the mean. The x-axis of DSE depolarization response curves are log-scale seconds of the duration of the depolarization used to elicit DSE. Depolarization response curves are obtained to determine pharmacological properties of endogenous 2-AG signaling by depolarizing neurons for progressively longer durations (50 ms, 100 ms, 300 ms, 500 ms, 1 s, 3 s, and 10 s).

### 2.3. Flamindo cAMP Assay

#### 2.3.1. Cell Culture and Transfection

HEK293 cells were purchased from ATCC. Cells were cultured in high glucose Dulbecco’s Modified Eagle Medium (Thermo Fisher Scientific, Waltham, MA, USA) and supplemented with 10% fetal bovine serum and a 1% Pen/Strep solution. Cultures were maintained at 37 °C with an atmosphere of 5% CO_2_. For the imaging experiments, the cells were dissociated using trypsin-EDTA (0.05%) and cultured on poly-d-lysine pre-coated 18 mm glass coverslips in 12-well plates. One day post-plating, the cells were transfected with the receptor of interest (rat CB_1_), the fluorescent protein EYFP, and the Pink Flamindo cAMP indicator [23], using Lipofectamine 2000 Transfection Reagent (Thermo Fisher Scientific). After 3.5 h, the transfection reagent was replaced with cell culture media and the cells used for experiments within two days of transfection.

#### 2.3.2. Cell Imaging and cAMP Binding Assay

Transfected HEK293 cells, were imaged in an extracellular solution containing (mM) NaCl 119, KCl 5, CaCl_2_ 2, MgCl_2_ 1, glucose 30 and HEPES 20, using a Leica TCS SP5 confocal microscope with an oil-immersion 20× objective. Images were acquired using an argon (40%), DPSS 561, with fluorescent wavelength settings set to 488–550 nm (EYFP), and 594–773-nm (Pink Flamindo). Drugs were initially prepared as a stock in DMSO or ethanol, then diluted using extracellular solution to their final concentration shortly before use.

Pink Flamindo is a fluorescent protein cAMP-indicator where increasing magnitudes of brightness in expressing cells, is indicative of elevated levels of cellular cAMP. CB1 agonists inhibit cAMP accumulation. CB1-transfected HEK293 cells were prepared as described above and were used to measure the inhibition of forskolin (Fsk)-induced production of cAMP, caused by the CB1 agonist, 2-AG (2.5 μM). The test compound and 2-AG were co-applied, followed by the potent adenylyl cyclase activator, forskolin (Fsk; 100 μM). Images were acquired every 30 s for 15 min and then analyzed using FIJI software with the 1-click ROI manager plugin [24], to measure the change in fluorescence intensity. Target cells were chosen by taking the first image in the series, increasing the brightness, and marking cells that exhibited a baseline Pink Flamindo fluorescence. Occasional (<5%) cells exhibited a high baseline fluorescence relative to the general transfected cell population. These cells were excluded from analysis since they were close to saturation. This mask of identified cells (typically 15–25 per experiment) was then applied to the image series. Baseline fluorescence intensity was normalized to 100 based on the first two minutes of the time series.

### 2.4. Methods for Dorsal Root Ganglion Cell Culture

DRGs were harvested from P-0 through P-14 day old rat pups following strict IACUC guidelines for the ethical care and use of laboratory animals. Rats were euthanized using isoflurane inhalation and cervical dislocation. DRGs were harvested using the protocol described by Sleigh et al. [25]. Briefly, rats were sprayed with 70% ethanol and the dorsal side was opened along the longitudinal axis with surgical scissors. The spine was removed, cleaned of excess muscle, and cut longitudinally along the dorsal and ventral surfaces. It was then placed into cold Dissection Solution (Earl’s Balanced Salt Solution (Gibco, 24010043), 10 mM MgCl_2,_ 1X GlutaMAX (Gibco, 35050061), Penicillin/ Streptomycin (500 μg/mL, Gibco, 15140122), and 10 mM HEPES (Thermo Fisher Scientific, BP310-1), and the spinal cord was carefully removed and discarded. DRGs were pulled from the vertebrate and placed into a 15 mL conical containing ice-cold Dissection Solution. DRGs were centrifuged at 100× *g* at 4 °C and the media was replaced with Dissection Solution containing 10 mg/mL collagenase type II (Gibco, 17101015). DRGs were incubated at 37 °C for 20 min. Trypsin/EDTA (Gibco, 15090046) was then added to a final concentration of 0.05% and the tissue was incubated a further 3 min. Tissue was centrifuged at 4 °C as above and washed three times in ice-cold DMEM (Gibco, 11965126) containing 10% fetal calf serum. DRGs were then triturated 30 times in 4 mL of this medium, centrifuged, and resuspended in 3 mL of cold culture medium (Neurobasal A (Gibco, 10888022), 2.5 mg/mL insulin, 5 mg/mL transferrin, 5 mg/mL nerve growth factor-b (Sigma, SRP4304), 1X B27 (Gibco, 17504044), 1X GlutaMAX, 10% Fetal calf serum). Cells were then counted and plated at a concentration of ~5000 cells/cm^2^ on coverslips coated with poly-d lysine (Sigma-Aldrich, St. Louis, MO, USA, P-7886), and laminin (Millipore, Burlington, VT, USA, SCR127). Cells were cultured at 37 °C and 5% CO_2_ with one half of the media changed every 3–4 days.

### 2.5. Methods for Calcium Imaging

DRGs were treated with Fluo4-AM (5 μM) for 30 min at 37 °C after which the cells were washed in extracellular solution (see electrophysiology) for 20 min to allow for de-esterification of Fluo4-AM. Fluorescence was then monitored on a Nikon TE200 inverted microscope (Nikon Instruments, Melville, NY, USA) with a 10× objective, a Hamamatsu Photonics (Hamamatsu City, Japan, Flash 4.0 camera and Nikon Elements AR software, (version 4.50, Nikon Instruments, Melville, NY, USA) which controlled a Spectra X light engine (Lumencor Inc., Beaverton, OR, USA) for stimulation of fluorescence. Target DRGs were chosen based on neuronal morphology using a Brightfield image acquired before the experiment. This mask was then applied to the image series. Images were acquired every 30 s for 15 min and then analyzed using FIJI software with the 1-click ROI manager plugin [24], to measure the change in fluorescence intensity over 15 min. Baseline fluorescence intensity was normalized to zero based on the two minutes preceding drug application.

### 2.6. Statistics

For electrophysiology experiments, the data were fitted with a nonlinear regression (Sigmoidal dose response; GraphPad Prism 6, La Jolla, CA, USA), allowing calculation of an ED50, the effective dose or duration of depolarization at which a 50% inhibition is achieved. A statistically significant difference between these curves is defined as non-overlapping 95% confidence intervals of the ED50s. Values on graphs are presented as mean ± S.E.M. Comparisons of the effects of various THCV concentrations were made using a one-way ANOVA with Dunnett’s post hoc vs. control. Statistical comparisons of single drug effects (e.g., baclofen alone vs. baclofen with CBDV) were conducted using an unpaired *t*-test.

For the cAMP assay, we used an area under the curve (AUC) analysis for time points from 0 to 15 min. Administration of a drug concentration series allowed the calculation of an IC50 for THCV in this system using GraphPad Prism 6. For a given experimental treatment, a same-day control forskolin-only experimental control was included. Experimental results were compared to their respective same-day controls.

### 2.7. Drugs

The drugs CBC, CBDA, CBDV, and 2-AG were purchased from Cayman Chemical (Ann Arbor, MI, USA), and CBDVA and THCV were purchased from Cerilliant Corporation (Round Rock, TX, USA).

## 3. Results

### 3.1. CBC Modestly Inhibits CB1 Signaling in Autaptic Hippocampal Neurons while CBDA, and CBDVA Are without Effect

Cannabichromene (CBC) is frequently cited as a phytocannabinoid with attractive properties [10]. Some interactions and similarities between CBC and Δ^9^-THC were described in the early eighties [26,27]. CBC produced mild hypothermia in mice and affected motility in electroshock-induced model of seizures, but only at very high doses (75 mg/kg). Marketing for CBC-containing products often cite studies reporting anti-inflammatory [28] and analgesic [27] properties that can be mediated through CB2 receptors at which CBC has been described as more potent agonist than Δ^9^-THC [29]. CBC has been shown to activate the transient receptor potential ankyrin type-1 (TRPA1) receptor at a relatively low concentration (EC50 = 90 nM) [30] and produced antinociceptive effects in rats following brainstem injection of low nanomole doses [31]. Other potential mechanisms of action include direct interaction with CB1 receptors, either at orthosteric or allosteric sites, and altered synthesis/metabolism of endocannabinoids. Although CBC has an impact on CB1-related behavior in mice, the effect is only prominent at high doses (100 mg/kg) and is not reversed by the CB1 inverse agonist SR141716 [32]. We tested CBC at 1 μM, a concentration chosen for CBC and other phytocannabinoids because it represents a likely physiological ceiling concentration that consumers might encounter (discussed in [13]). It is also a concentration that has been shown to affect the activity of extracellular signal-regulated kinases 1 and 2 (ERK1/2) and viability of neuronal stem cells [33]. At 1 μM, CBC did not alter excitatory post-synaptic current (EPSC) amplitudes (Figure 2A, EPSC charge relative to baseline (1.0 = no effect) CBC: 1.04 ± 0.02, *n* = 4; *p* = 0.23 by one-sample *t*-test vs. baseline 1.0), indicating that CBC does not directly alter excitatory neurotransmission in this system.

To test whether CBC modulated cannabinoid signaling, we tested for its effects on depolarization-induced suppression of excitation (DSE), a form of endogenous 2-AG- and CB1-mediated retrograde signaling present in autaptic hippocampal neurons. As described in the methods section, successively longer depolarizations (100 ms, 300 ms, 500 ms, 1 s, 3 s, and 10 s) result in greater inhibition of EPSCs, yielding a ‘depolarization dose-response’ curve. A potentiator of cannabinoid signaling would be expected to shift this curve to the left, as is the case with positive allosteric modulators [34]. Conversely, an inhibitor of cannabinoid signaling would be expected to shift this curve to the right and depress maximal DSE, as seen with negative allosteric modulation [35]. We found that CBC did not alter the EC50 for DSE at 1 μM (Figure 2B, Table 1).

Cannabidiolic acid (CBDA) is the acidic precursor of CBD [36]. Although a U.S. patent was written, citing CBDA as a possible treatment for autism [37], and it was reported to have anti-nausea properties [38], little is known about its pharmacology. In our model at 1 μM, we found that CBDA did not alter EPSC amplitudes (Figure 2C, 0.99 ± 0.03, *n* = 5; *p* = 0.69 by one-sample *t*-test vs. baseline 1.0). We also did not see a significant change in DSE responses (Figure 2D, Table 1).

Cannabidivarinic acid (CBDVA) is the acidic precursor to CBDV and has received little attention until recently. CBDVA has a high oral bioavailability [39]; however, it seems to have poor brain penetration [40]. CBDVA was reported to inhibit DAGLα, however to a lesser extent than CBDV and CBDA [30]. We found that CBDVA did not alter excitatory neurotransmission at 1 μM (Figure 2E, 1 μM CBDVA: 1.02 ± 0.01, *n* = 5; *p* = 0.10 by one-sample *t*-test vs. baseline 1.0) and did not significantly alter DSE (Figure 2F, Table 1).

### 3.2. THCV Potently Inhibits CB1 Signaling

Tetrahydrocannabivarin (THCV) is a homolog of Δ^9^-THC where, like CBDV, the lipophilic side chain is shortened by two methylene bridges [41]. THCV was reported to act as a competitive antagonist at CB1 receptors [42] and as an agonist at higher concentrations [43]. THCV was shown to have both anti-convulsant [44] and anti-inflammatory [38] properties, consistent with the findings by Thomas et al. [42]. While THCV did not alter neurotransmission on its own (Figure 3A, EPSC charge relative to baseline after THCV (1 μM): 1.01 ± 0.02, *n* = 5, *p* = 0.70 by one-sample *t*-test vs. baseline 1.0), THCV inhibited DSE in a concentration-dependent manner. The effect of THCV was surprisingly potent: 100 nM THCV was sufficient to fully block DSE in response to a 10 s depolarization and even 100 pM THCV significantly reduced DSE (Figure 3B,C; Table 2). The calculated IC50 for THCV in this system was 708 pM. The range of concentrations over which THCV acted was unusually long and a Schild plot yielded a slope of 0.52 (Figure 3D), potentially an indication of a non-competitive antagonism, negative cooperativity, or a second target in this system.

Though Thomas et al. [42] reported that THCV is a competitive antagonist at CB1, DSE signaling occurs because 2-AG is synthesized postsynaptically, crosses the synaptic cleft, and acts at CB1 presynaptically. In principle, the effect of THCV on DSE might occur either presynaptically (i.e., at CB1 signaling) or post-synaptically (at some aspect of 2-AG production or transport). If the effect is pre-synaptic, then THCV should also inhibit bath-applied 2-AG. We tested this by applying 500 nM 2-AG and attempting to reverse 2-AG inhibition of EPSCs by switching into 2-AG + THCV (100 nM). We found that 100 nM THCV readily reversed inhibition by 500 nM 2-AG (Figure 3E,F, relative EPSC charge after 2-AG (500 nM): 0.32 ± 0.05; after 2-AG + THCV (100 nM): 0.79 ± 0.06; *n* = 5, *p* < 0.01 by paired *t*-test). Given then that THCV is acting presynaptically, it may be acting at CB1, but it might also be interfering more generally with presynaptic G_i/o_ signaling. To test this possibility, we attempted to block the inhibition of EPSCs by GABA_B_ agonist baclofen (25 μM), since activation of the GABA_B_ receptor also inhibits EPSCs via the G_i/o_ pathway in these neurons [45]. We found that baclofen responses were unimpeded by the presence of 100 nM THCV (Figure 3G,I; Relative EPSC charge after baclofen (25 μM) applied in presence of THCV (100 nM): 0.25 ± 0.09, *n* = 3; baclofen only: 0.21 ± 0.03, *n* = 3).

Given the potency of THCV to antagonize CB1 signaling during DSE, we tested for the effect of THCV on cAMP signaling to learn whether THCV would be similarly potent in other signaling pathways. CB1 activation is well known to inhibit the activity of adenylyl cyclase [46,47]. Using HEK293 cells transfected with CB1 and the cAMP indicator Pink Flamindo [23], we tested forskolin-induced changes in cAMP levels in response to 2-AG (2.5 μM) alone or co-treatment with various concentrations of THCV. The 2-AG reduces cAMP accumulation, an effect that is concentration-dependently inhibited by THCV with an IC50 of 7.3 nM (Figure 4A,B). THCV inhibition of 2-AG suppression of neurotransmitter release is therefore ~10× more potent than inhibition of cAMP accumulation in HEK293-CB1 cells.

### 3.3. CBDV Inhibits Endocannabinoid Signaling Postsynaptically 

Cannabidivarin (CBDV) is a homolog of CBD where the lipophilic side chain is shortened by two methylene bridges [48]. CBDV has been shown to have anti-convulsant properties [49] though perhaps independently of CB1 [50]. CBDV activates and desensitizes TRPV1 transient receptor potential cation channel subfamily V member 1 (TRPV1) with promising anti-epileptic implications [51]. Moreover CBDV suppresses the expression of epilepsy-related genes following chemical convulsant treatment [52], suggesting it may be useful in preventing the development of epilepsy. CBDV has been the subject of preclinical studies for the treatment of epilepsy [53] and has been shown to rescue cognitive deficits and motor defects in a mouse model of Rett syndrome [54].

CBDV did not directly alter neurotransmission when applied (Figure 5A, CBDV 1 μM: 1.01 ± 0.02, *n* = 5, *p* = 0.40 by paired *t*-test). However, CBDV inhibited DSE at 1 μM but not at 100 nM (Figure 5B, Table 1). A sample trace is presented in Figure 5C.

To explore the mechanism of CBDV action further, we tested the effect of CBDV on responses to bath-applied 2-AG. As noted for THCV, inhibition of DSE may occur due to altered CB1 signaling but also as a consequence of altered 2-AG production. If the effect of CBDV was due to inhibition of CB1 signaling, then CBDV should similarly inhibit the effects of bath-applied 2-AG. However, we did not see an inhibition of 2-AG responses, indicating that CBDV may act post-synaptically to impact 2-AG availability (Figure 5D,E: relative EPSC charge after 2-AG (500 nM): 0.56 ± 0.08; after 2-AG + CBDV (1 μM); 0.55 ± 0.09, *n* = 5, NS by paired *t*-test).

### 3.4. CBC, CBDA, CBDVA Do Not Alter Calcium Responses in Dorsal Root Ganglion Neurons

Several groups have reported that phytocannabinoids activate transient receptor potential (TRP) receptors (reviewed in [15]), though the effects often require concentrations in excess of 10 μM (e.g., [51]). Some however report responses at low-micromolar concentrations [30]. TRP receptors are ion channels that are opened by different stimuli that include chemicals but also temperature. We tested the activity of these phytocannabinoids in a second neuronal model, dorsal root ganglion neurons (DRGs) cultured from a rat. These neurons are known to natively express a variety of TRP channels, including TRPV1, TRPV3, TRPV4, and TRPA1, which have been reported to be activated by phytocannabinoids [30]. DRG subtypes differentially express these channels, underscoring one challenge in working with DRGs: they are not a uniform neuronal population. Several efforts have been made to classify DRG subpopulations [55,56]. We used calcium imaging to permit visualization of calcium influx of multiple neurons in response to TRP channel activation. We tested the TRPV1 agonist capsaicin (1 μM), finding that it activated ~40% of DRGs, consistent with reported literature (e.g., Figure 6A and Figure 7A, [15]). According to De Petrocellis et al., THCV was one of the most potent agonists of TRPV1, with an EC50 of 1.5 μM, and one of the highest efficacies reported out of a dozen phytocannabinoids tested [30]. THCV proved to be the most likely to elicit a response, but most (94%) cells did not respond to THCV (Figure 6A). Those cells that did (6%), saw desensitizing (Figure 6B) or sustained (Figure 6C) responses, but these were infrequent (eight for each type of response (3%) of 249 cells). Significantly, most cells that responded to capsaicin failed to respond to THCV (e.g., Figure 6C). This suggests that the THCV-induced calcium responses that were observed were not due to TRPV1 activation by THCV. According to De Petrocellis, CBDV was also reported to activate TRP receptors [30], with an EC50 of 3.5 μM (for TRPV1), but their follow-on study using electrophysiological measurements only found effects at concentrations at 10 μM or higher [51]. We found that CBDV seldom (7%) activated calcium responses even in cells that were strongly activated by capsaicin (Figure 6D). On rare occasions, some cells appeared to see an increase in the frequency of spontaneous calcium transients (5 out of 158 cells Figure 6E) or activation of a steady calcium current (6 out of 158 cells Figure 6F). However, these responses were infrequent (3–4%).

Of the remaining phytocannabinoids, CBDA (1 μM), on rare occasions (2%), elicited a sustained calcium response (2 out of 126 cells, Figure 7A,B). Similarly, CBC elicited a brief calcium response in a single cell out of 99 tested (Figure 7C,D). CBDVA never elicited significant calcium responses (Figure 7E).

## 4. Discussion

There has been increasing interest in ‘minor’ phytocannabinoids due to the changing legal landscape and advances in their synthesis and extraction. These compounds are now being introduced in consumer products and are marketed as having health benefits; however, they remain largely uncharacterized. We evaluated a panel of phytocannabinoids—CBC, CBDA, CBDV, CBDVA, and THCV—using two neuronal models. Autaptic hippocampal neurons express an endogenous CB1/2AG-based retrograde form of synaptic plasticity, while DRGs natively express a variety of TRP channels. Our chief findings show that three of the five compounds tested—CBC, CBDA, and CBDVA—had a slight or no effect in either model. However, CBDV and THCV each inhibited cannabinoid signaling, albeit via distinct mechanisms. THCV antagonized CB1 signaling as reported by others [42], but with unusually high potency, inhibiting endogenous 2-AG/CB1 signaling at concentrations as low as 100 pM. In our most striking finding, CBDV did not directly inhibit the CB1 receptor, but instead acted postsynaptically, perhaps by interfering with 2-AG production. In DRGs, despite several reports that phytocannabinoids activate calcium-permeable TRP channels, such as TRPV1, most compounds induced only infrequent, if any, calcium responses at 1 μM. These rare responses were therefore inconsistent with minor cannabinoid activation of TRPV1, for example, which is expressed in a large fraction of DRG neurons.

We chose 1 μM concentrations to test these compounds because this rests at the high end of the concentration range in which an individual is likely to encounter. We previously discussed this for the examples of THC [17] and CBD [13]. For example, Dravet Syndrome patients achieve ~1 μM blood plasma concentrations after 20 mg/kg CBD/Epidiolex treatments [57]. The pharmacokinetic properties of these compounds may nonetheless vary substantially and impact the final concentration and effect of a given phytocannabinoid [58,59]. Though minor cannabinoids are found at low concentrations in the cannabis plant, purification and synthesis of these compounds allow their incorporation into products to deliver doses comparable to THC and CBD. Nonetheless, it is unlikely that the minor cannabinoids that were inactive in these two model systems have activity at CB1 receptors or TRP channels in human neurons.

The THCV findings are interesting in several respects. THCV was reported to produce hypophagia in both non-fasted and fasted mice at doses of 3 mg/kg [60], possibly mediated by CB1 antagonism. A study in healthy volunteers showed that THCV increases neural response to rewards and aversive stimuli connected with food [61], while another report indicated that antagonism by THCV of CB1 signaling seems to be free of adverse events associated with CB1 inverse agonists, such as rimonabant [62]. As much of the interest in phytocannabinoids has to do with their CNS effects, the high potency of THCV in a neuronal model is significant.

The minor cannabinoids have been dismissed by some because their concentrations in the plant are relatively low. However, if a compound that is present in cannabis at 1% of the concentration of THC is 100-fold more potent, then the contribution of this compound to the net effect of cannabis may be significant and must be considered. This point also ties into the proposed ‘entourage effect’ [63]. The entourage effect generally refers to synergistic action by compounds present with THC in cannabis. In the case of THCV, the presumed net effect would be to oppose the action of THC at CB1 and may contribute to the net effect of cannabis in cultivars that have higher levels of THCV. Notable also was the broad distribution of THCV concentrations that affected neuronal cannabinoid signaling. A Schild analysis of these data is consistent with negative cooperativity between THCV and CB1 in this system or the possibility that THCV acts on a second target.

Our most novel finding is that CBDV interferes with cannabinoid signaling, not by inhibiting CB1 receptors, but postsynaptically, perhaps by hindering the production of the endocannabinoid 2-AG. However, the underlying mechanism for this remains to be elucidated and can be investigated in future studies.

Our negative findings for CBDA and CBDVA in autaptic neurons do not rule out activity of these compounds at other components of the cannabinoid signaling system. This includes other receptors [64], enzymes such as ABHD6 and ABHD12 [65], and members of the TRP family of ion channels not expressed in DRG neurons, several of which are modulated by endocannabinoids [16,66]. Several of the phytocannabinoids tested are entering into clinical trials based on their proposed health effects.

Our study of TRP responses in DRGs yielded mostly negative results. This can likely be attributed to two factors, one being the higher concentrations employed by most studies that have reported effects, and the second due to differences in expression systems versus a natively expressing neuronal population. Our results for CBDV and TRPV1 are in agreement with [51], who saw no effects for CBDV at concentrations below 10 μM using electrophysiological measures in TRPV1-expressing HEK293 cells. However, based on the finding by De Petrocellis et al. [30] that THCV activated TRPV1 with an EC50 of 1.5 μM and an efficacy of >60% of ionomycin, one would have expected a TRPV1 response in ~40% of DRG neurons. The difference may lie in the assay employed, which relied on responses in a homogenized sample and was therefore an additional step removed from an intact expression system. While we did observe occasional responses to phytocannabinoids, they were prohibitively infrequent to permit further investigation.

In summary, we found that, in a sampling of five phytocannabinoids that have attracted general interest in the population, two exerted substantial effects on CB1 and 2-AG-mediated cannabinoid signaling in a neuronal model. THCV and CBDV both inhibited cannabinoid signaling. However, while THCV acted as a CB1 antagonist, CBDV acted postsynaptically to inhibit DAGLα-mediated 2-AG production. These findings highlight the importance of testing phytocannabinoid interaction with all components of the cannabinoid signaling system; moreover, the remaining ‘minor’ phytocannabinoids may offer more interesting surprises.

## Figures and Tables

**Figure 1 molecules-26-05352-f001:**
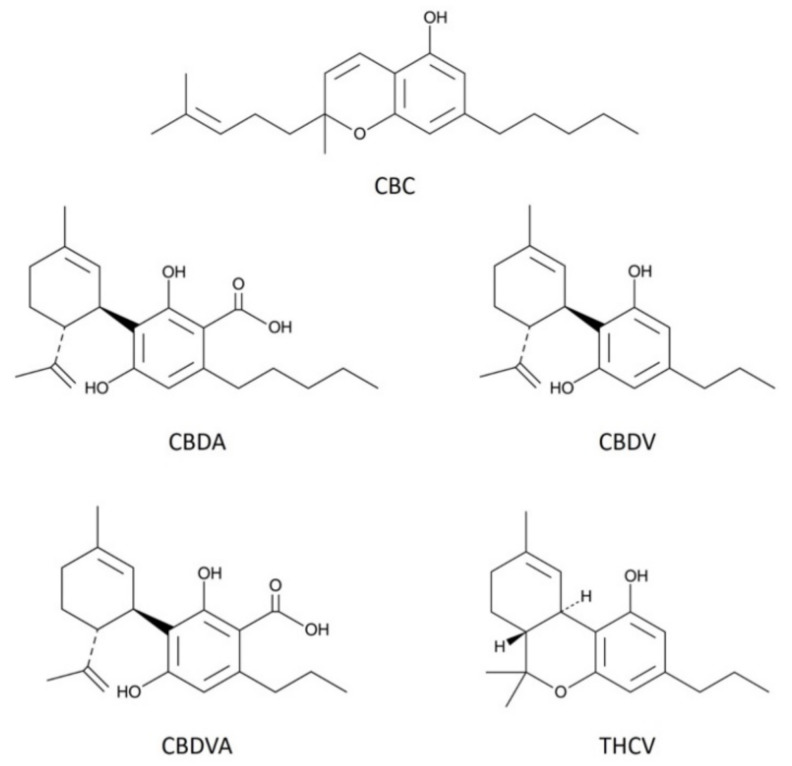
Structures of phytocannabinoids examined in the current study. The figure shows chemical structures of the five phytocannabinoids used in this study; cannabichromene (CBC), cannabidiolic acid (CBDA), cannabidivarin (CBDV), cannabidivarinic acid (CBDVA), and Δ^9^-tetrahydrocannabivarin (THCV).

**Figure 2 molecules-26-05352-f002:**
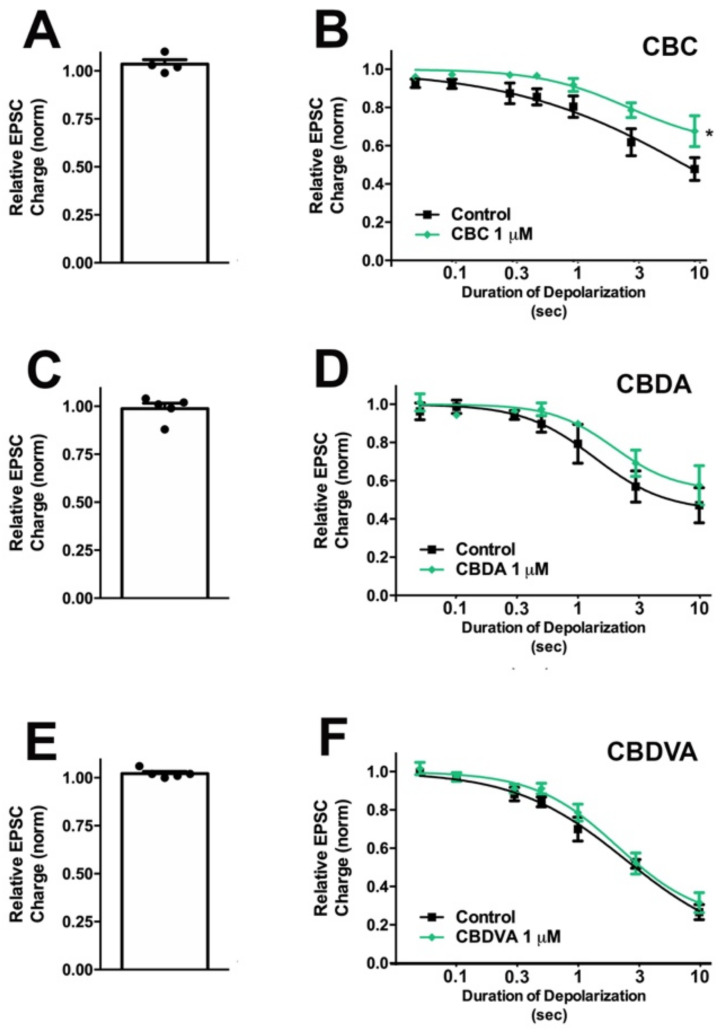
CBC modestly inhibits CB1 signaling in autaptic hippocampal neurons, while CBDA and CBDVA are without effect. (**A**) CBC, (**C**) CBDA, and (**E**) CBDVA have no direct effect on EPSCs. (**B**) CBC modestly inhibits maximal DSE. (**D**,**F**) CBDA and CBDV do not have a significant effect on DSE-mediated inhibition of EPSCs. *, *p* < 0.05, paired *t*-test for 10 s inhibition, drug vs. baseline.

**Figure 3 molecules-26-05352-f003:**
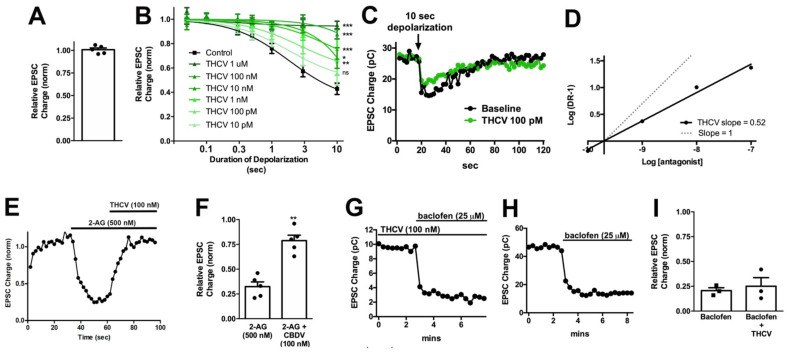
THCV potently inhibits presynaptic CB1 responses in autaptic neurons. (**A**) THCV (1 μM) has no direct effect on EPSCs. (**B**) THCV concentration-dependently reduces DSE inhibition of EPSCs, with significant effects even at 100 pM. *, *p* < 0.05, ***, *p* < 0.005 one-way ANOVA with Dunnett’s post hoc vs. control. (**C**) Sample DSE responses before and after treatment with 1 μM THCV. (**D**) A Schild analysis shows that the Schild slope is less than 1. (**E**) Sample time course showing reversal of 2-AG inhibition by THCV (100 nM). (**F**) Summarized data showing that THCV reverses 2-AG action. **, *p* < 0.01 by paired *t*-test. (**G**,**H**) Sample time courses showing that baclofen (25 μM) responses are similar with and without pre-treatment with THCV (100 nM). (**I**) Summarized data for baclofen/THCV vs. baclofen alone. *p* > 0.05 unpaired *t*-test.

**Figure 4 molecules-26-05352-f004:**
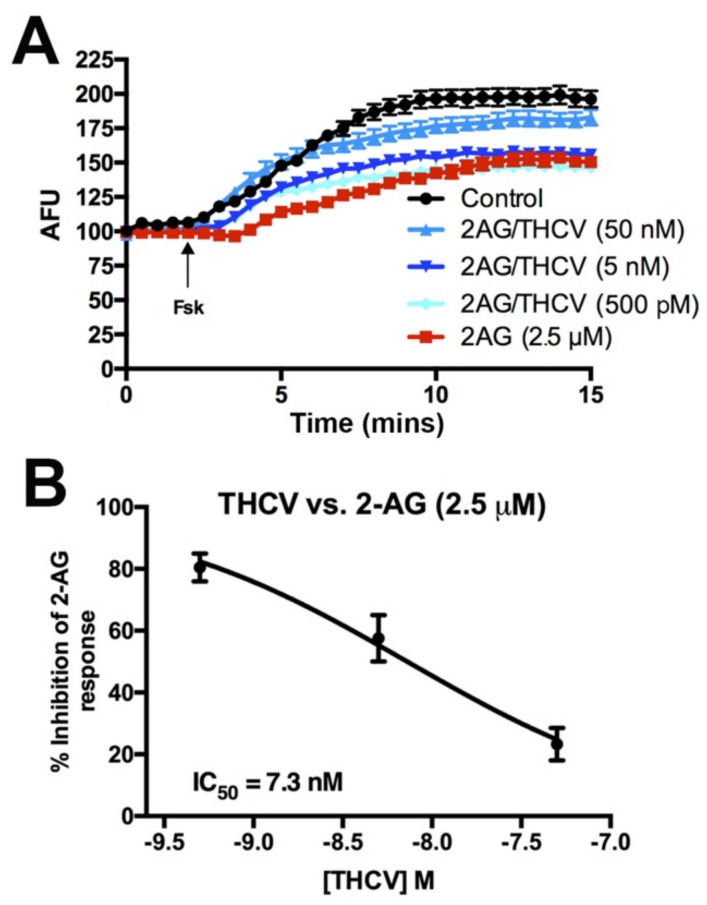
THCV inhibits 2-AG-mediated inhibition of adenylyl cyclase, but less potently than inhibition of neurotransmission. (**A**) Sample time courses from one set of experiments showing effects of drug combinations on forskolin-induced increases in cAMP in HEK293 cells transfected with mCB1 and the pink Flamindo cAMP indicator. (**B**) Summary cAMP responses show a concentration-dependent inhibition of 2-AG, with an IC50 of 7.3 nM.

**Figure 5 molecules-26-05352-f005:**
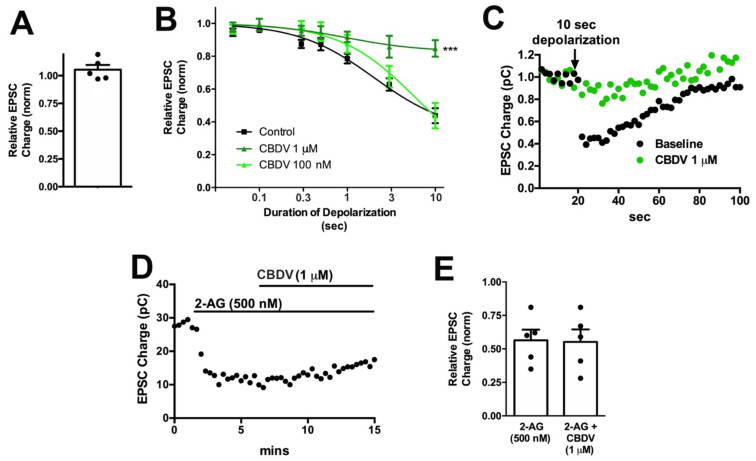
CBDV inhibits DSE post-synaptically in autaptic neurons.; (**A**) CBDV (1 μM) has no direct effect on EPSC; (**B**) CBDV blocks DSE at 1 μM but not at 100 nM; (**C**) sample DSE responses before and after treatment with 1 μM CBDV; (**D**) sample time course showing non-reversal of 2-AG inhibition by CBDV (1 μM). (**E**) Summarized data showing that CBDV (1 μM) does not reverse the effect of 2-AG (500 nM). ***, *p* < 0.001, one-way ANOVA with Dunnett’s post-hoc test.

**Figure 6 molecules-26-05352-f006:**
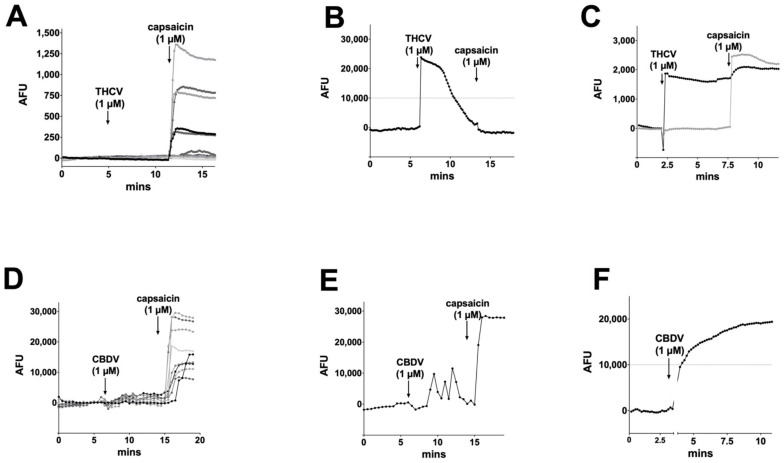
THCV and CBDV calcium responses in DRGs. (**A**) THCV (1 μM) rarely induced a calcium response in DRGs, while the TRPV1 agonist capsaicin (1 μM) induced responses in a large subset of DRGs. (**B**) In a small number of cells (~3%), THCV induced a desensitizing current. (**C**) A few cells (~3%) showed sustained responses to THCV. (**D**) CBDV (1 μM) did not typically induce a calcium response in DRGs. (**E**) In a few cells (~3%), CBDV appeared to increase spontaneous Ca transients. (**F**) A few cells (~4%) had a sustained calcium response to CBDV. AFU, arbitrary fluorescence units.

**Figure 7 molecules-26-05352-f007:**
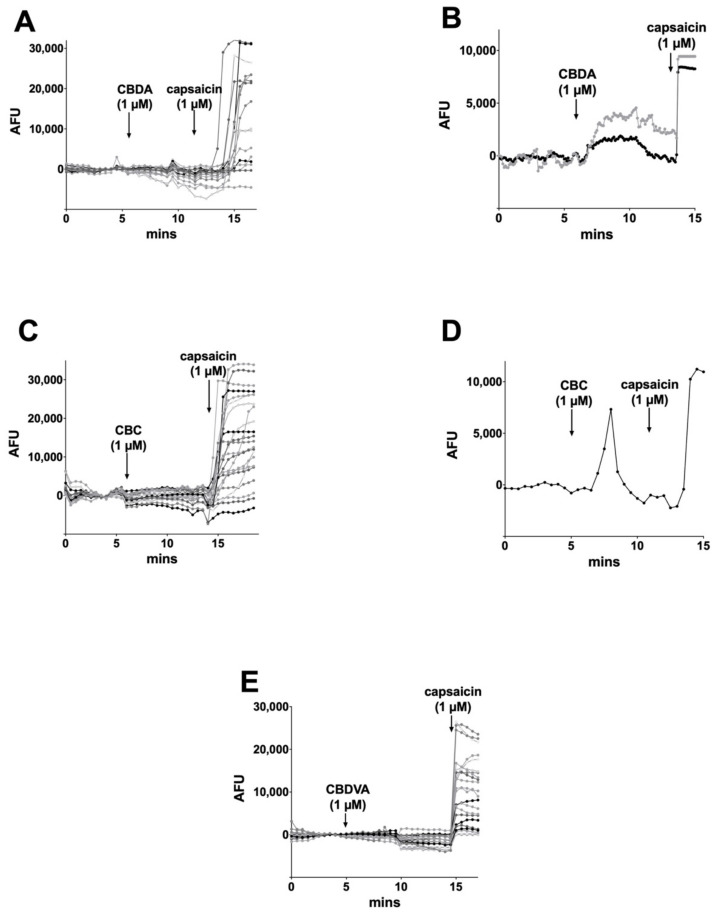
CBDA, CBC, and CBDVA rarely increased intracellular calcium in DRGs. (**A**) Sample time course shows calcium response in DRGs after treatment with CBDA (1 μM) followed by TRPV1 agonist capsaicin (1 μM), (**B**) on rare occasions (2%) DRG neurons responded to CBDA, (**C**) Sample time course shows calcium response in DRGs after treatment with CBC (1 μM) followed by capsaicin (1 μM), (**D**) on rare occasions (~1%) brief responses were seen after CBC treatment. (**E**) CBDVA failed to induce calcium responses in any cells tested.

**Table 1 molecules-26-05352-t001:** Phytocannabinoid responses in autaptic hippocampal neurons. Values for EPSC inhibition in response to longest depolarization (10 s) for baseline and given phytocannabinoid. Paired *t*-test was used to compare maximal inhibition at 10 s depolarization vs. baseline in a given cell. Effective dose 50 (ED50, with 95% confidence interval) for depolarization-response curves indicating duration of depolarization that gave a 50% maximal response for the phytocannabinoids tested. None of the phytocannabinoids significantly altered the ED50.

	Inhibition at 10 s Depolarization				ED50 (95% CI)		
	Concentration	Control	Drug	Significant	*p* Value	Control	Drug
CBC	1 μM	0.48 ± 0.06	0.68 ± 0.08	Yes	0.01	1.58 s (0.78–3.18)	3.90 s (1.40–10.87)
CBDA	1 μM	0.47 ± 0.09	0.58 ± 0.10	No	0.11	1.72 s (0.84–3.54)	3.82 s (1.29–11.32)
CBDVA	1 μM	0.27 ± 0.04	0.32 ± 0.05	No	0.24	2.23 s (1.54–3.22)	3.10 s (1.93–4.99)
CBDV	100 nM	0.37 ± 0.05	0.44 ± 0.08	No	0.17	1.78 s (1.19–2.66)	6.05 s (2.32–15.72)
	1 μM	0.49 ± 0.07	0.85 ± 0.05	Yes	0.0013	1.78 s (1.19–2.66)	0.97 s (0.18–5.19)
THCV	100 nM	0.39 ± 0.04	0.88 ± 0.04	Yes	0.0018	1.84 s (1.20–2.82)	ambiguous

**Table 2 molecules-26-05352-t002:** THCV potently inhibits DSE in autaptic hippocampal neurons. Values for EPSC inhibition in response to longest depolarization (10 s) for baseline and THCV at various concentrations (10 pM–1 μM). One-way ANOVA with Dunnett’s post hoc test was used to compare maximal inhibition at 10 s depolarization vs. controls. Effective dose 50 (ED50, with 95% confidence interval) for depolarization-response curves indicating duration of depolarization that gave a 50% maximal response for various concentrations of THCV.

	Concetration	Inhibition at 10 s	Significant	*p* Value	ED50 (95%CI)
**Control**	-	0.39 ± 0.04			1.84 s (1.20–2.82)
THCV	10 pM	0.56 ± 0.10	No	0.285	1.83 s (0.75–4.47)
	100 pM	0.66 ± 0.10	Yes	0.011	2.68 s (0.76–9.43)
	1 nM	0.75 ± 0.07	Yes	0.0008	7.35 s (1.15–47)
	10 nM	0.68 ± 0.08	Yes	0.014	ambiguous
	100 nM	0.87 ± 0.04	Yes	<0.0001	ambiguous
	1 μM	0.95 ± 0.03	Yes	<0.0001	ambiguous

## Data Availability

The data presented in this study are available on request from the corresponding author.

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
