# Peer review of "An Evaluation of Understudied Phytocannabinoids and Their Effects in Two Neuronal Models"

_molecules, 2021, doi:10.3390/molecules26175352_

Round 1

Reviewer 1 Report

I have revised the work entitle: An evaluation of understudied phytocannabinoids and their effects in two neuronal models. It is an interesting work, however, I have some suggestions:

1.- It is not clear why authors select these five phytocannabinoids, it is due to there are more than 100. It is possible that after the THCD and CBD, these five selected are the most found in cannabis, however, it is possible the presence of others with highest potency. 

2.- Due to from THCB and CNB are the most studied, these could be interested to be used as drug-drug interactions to verify their possible site binding competition. Also, these ligands are requiere to be used as positive controls.

3.- I suggest to authors employ docking studies to explore the CB1 and CB2 recognition  properties to be then compared and discussed with the obtained experimental results. 

Author Response

1.- It is not clear why authors select these five phytocannabinoids, it is due to there are more than 100. It is possible that after the THCD and CBD, these five selected are the most found in cannabis, however, it is possible the presence of others with highest potency. 

A: This project was intended to be an initial survey with the hope that we will be able to follow up with studies.   That said, it was not an easy choice to select five phytocannabinoids from the many possible compounds.  In page 2 of the introduction we described these compounds as being among the “more widely promoted” phytocannabinoids.  This was intended to allow for a variety of compounds reported to have different effects.   We have added a sentence to the introduction that more clearly outlines our rationale for choosing these compounds. 

2.- Due to from THCB and CNB are the most studied, these could be interested to be used as drug-drug interactions to verify their possible site binding competition. Also, these ligands are requiere to be used as positive controls.

A:  We were obliged to choose a limited number of phytocannabinoids for our initial survey.  THCB and CNB are certainly very interesting and would make excellent targets for a follow-on study. 

3.- I suggest to authors employ docking studies to explore the CB1 and CB2 recognition  properties to be then compared and discussed with the obtained experimental results. 

A: While docking studies would be interesting, the focus of the manuscript was on the actions of these compounds in two neuronal models.  Docking studies, while interesting, would be better suited to a follow-on study.   Such a study could also investigate actions at CB2, which is not present in the model systems employed here and therefore not a subject of the current study.

Reviewer 2 Report

Manuscript is very interesting. It is clearly written  and carefully edited. I have only some minor suggestions:

Line 72-73 - unnecessary sentence in Introduction.

Line 74: it should be „Materials and methods”

Fig. 1 should be a part of 2.6. subsection. It should be also cited in text before appearance in manuscript.

Table 1-2: what does it mean symbols “@”?

Author Response

Line 72-73 - unnecessary sentence in Introduction.

Line 74: it should be „Materials and methods”

Fig. 1 should be a part of 2.6. subsection. It should be also cited in text before appearance in manuscript.

Table 1-2: what does it mean symbols “@”?

A: We have updated the manuscript according to these suggested changes.

Reviewer 3 Report

The submitted research article analyses the biological effects of 5 minor phytocannabinoids on cannabinoid signaling in two neuronal cell models. Two of the tested molecules, THCV and CBDV, affected the endogenous endonannabinoid signalling, but with different biological mechanism, pointing out the need of additional study on minor phytocannabinoids .  

The manuscript is well done and structured. I only have the following minor queries:

  1. 132 (Harada et al., 2017) 249 (Masahiro et. al., 1977) delete and insert the corresponding numbers from reference list; update reference list in case these are misssing references
  2. Give uniformity to the writing of cannabinoid receptors (i.e. CB1 or CB1)
  3. In methods Include the section "statistics"
  4. Include in the main text the reference to figure 1
  5. In general the summary of the biological properties and activities of "minor" phytocannabinoids is not a "result", but more suitable as background of the study and discussion points.
  6. 208"CBC modestly inhibits CB1 signaling in autaptic hippocampal neurons while CBDA, and CBDVA are without effect."255 "THCV potently inhibits CB1 signaling." 327 "CBDV inhibits endocannabinoid signaling postsynaptically by inhibiting DAGL ac" What do these sentenced represent? If they are paragraph titles, number paragraphs and follows the journal's style for paragraph titles.
  7. 271, 281 ".... Thomas et al. (2005) ...." include reference number

Author Response

  1. 132 (Harada et al., 2017) 249 (Masahiro et. al., 1977) delete and insert the corresponding numbers from reference list; update reference list in case these are misssing references
    1. A: The references have been updated.
  2. Give uniformity to the writing of cannabinoid receptors (i.e. CB1 or CB1)
    1. A: This is done.
  3. In methods Include the section "statistics"
    1. A: We have added a dedicated statistics section to the methods
  4. Include in the main text the reference to figure 1
    1. A: This is done.
  5. In general the summary of the biological properties and activities of "minor" phytocannabinoids is not a "result", but more suitable as background of the study and discussion points.
    1. A: With all due respect, we believe that given the diversity of phytocannabinoids, a brief introduction of the biological properties of a given phytocannabinoid just before showing the data for that compound serves as a more accessible introduction and rationale that makes the manuscript more readable.  
  6. 208"CBC modestly inhibits CB1 signaling in autaptic hippocampal neurons while CBDA, and CBDVA are without effect."255 "THCV potently inhibits CB1 signaling." 327 "CBDV inhibits endocannabinoid signaling postsynaptically by inhibiting DAGLa ac" What do these sentenced represent? If they are paragraph titles, number paragraphs and follows the journal's style for paragraph titles.          
    1. A: This formatting issue has been corrected.
  7. 271, 281 ".... Thomas et al. (2005) ...." include reference number
    1. A: This has been updated.